



GOVERNING EQUATIONS OF TRANSIENT SOIL WATER FLOW AND SOIL WATER
FLUX IN MULTI-DIMENSIONAL FRACTIONAL ANISOTROPIC MEDIA AND
FRACTIONAL TIME
M.L. Kavvas, A. Ercan, J. Polsinelli
Hydrologic Research Laboratory, Department of Civil & Envr. Engineering, University of
California, Davis, CA 95616, USA
ABSTRACT
In this study dimensionally-consistent governing equations of continuity and motion for
transient soil water flow and soil water flux in fractional time and in fractional multiple space
dimensions in anisotropic media are developed. Due to the anisotropy in the hydraulic
conductivities of natural soils, the soil medium within which the soil water flow occurs is
essentially anisotropic. Accordingly, in this study the fractional dimensions in two horizontal and
one vertical directions are considered to be different, resulting in multi-fractional multi-
dimensional soil space within which the flow takes place. Toward the development of the
fractional governing equations, first a dimensionally-consistent continuity equation for soil water
flow in multi-dimensional fractional soil space and fractional time is developed. It is shown that
the fractional soil water flow continuity equation approaches the conventional integer form of the
continuity equation as the fractional derivative powers approach integer values. For the motion
equation of soil water flow, or the equation of water flux within the soil matrix in multi-
dimensional fractional soil space and fractional time, a dimensionally consistent equation is also
developed. Again, it is shown that this fractional water flux equation approaches the
conventional Darcy's equation as the fractional derivative powers approach integer values. From
the combination of the fractional continuity and motion equations, the governing equation of
transient soil water flow in multi-dimensional fractional soil space and fractional time is
obtained. It is shown that this equation approaches the conventional Richards equation as the



fractional derivative powers approach integer values. Then by the introduction of the Brooks-
Corey constitutive relationships for soil water into the fractional transient soil water flow
equation, an explicit form of the equation is obtained in multi-dimensional fractional soil space
and fractional time. The governing fractional equation is then specialized to the case of only
vertical soil water flow and of only horizontal soil water flow in fractional time-space. It is
shown that the developed governing equations, in their fractional time but integer space forms,
show behavior consistent with the previous experimental observations concerning the diffusive
behavior of soil water flow.
INTRODUCTION
Various laboratory (Silliman and Simpson, 1987; Levy and Berkowitz, 2003) and field
studies (Peaudecerf and Sauty, 1978; Sudicky et al., 1983; Sidle et al., 1998) of transport in
subsurface porous media have shown significant deviations from Fickian behavior. As one
approach to the modeling of the generally non-Fickian behavior of transport, Meerschaert,
Benson, Baumer, Schumer, Zhang and their co-workers (Meerschaert et al. 1999, 2002, 2006;
Benson et al. 2000a,b; Baumer et al. 2005, 2007; Schumer et al. 2001, 2009; Zhang et al. 2007,
2008 and 2009) have introduced the fractional advection-dispersion equation (fADE) as a model
for transport in heterogeneous subsurface media. By theoretical and numerical studies the above
authors have shown that fADE has a nonlocal structure that can model well the heavy tailed non-
Fickian dispersion in subsurface media, mainly by means of a fractional spatial derivative in the
dispersion term of the equation. Meanwhile, they have also shown that fADE, with a fractional
time derivative, can also model well the long particle waiting times in transport in both surface
and subsurface environments. However, while the above-mentioned studies provided extensive
treatment of the fractional differential equation modeling of transport in fractional time-space by



subsurface flows, few studies have addressed the detailed modeling of the actual subsurface
flows in porous media in fractional time-space.

He (1998) seems to be the first scholar who proposed a fractional form of Darcy's equation

for water flux in porous media. Based on this fractional water flux equation, in his pioneering
work He (1998) then proposed a fractional governing equation of flow through saturated  porous
media. The left-hand-side (LHS) and the right-hand-side (RHS) of He's fractional Darcy flux
formulation have different units. As saturated flow equations, He's proposed governing equations
address the groundwater flow instead of the unsaturated soil water flow. Since the focus of our
study is soil water flow in fractional time-space, below we shall discuss the literature that
specifically addresses the fractional soil water flow equations.

As early as in 1960's Gardner and his co-workers (Ferguson and Gardner, 1963; Rawlins and

Gardner, 1963) questioned the classical diffusivity expression in the diffusion form of the
conventional Richards equation for soil water flow being only dependent on the soil water
content. Based on their experimental observations, they reported that diffusivity was also
dependent explicitly on time besides being dependent on the soil water content. Following on
these experimental observations, Guerrini and Swartzendruber (1992) hypothesized a new form
for Richards equation for horizontal unsaturated soil water flow in semi-rigid soils. Unlike the
assumption that the soil hydraulic conductivity K and soil water pressure head $\psi$ are only
dependent on the soil water content, they hypothesized that K and $\psi$ are also dependent
explicitly on time. This hypothesis led them to the formulation of the diffusivity coefficient D
within the diffusion form of the Richards equation as function of not only the soil water content
but also explicitly on time, that is $D = D(\theta,t) = E(\theta) t^m$ where E is a function of water content $\theta$
while m is a power value. The application of their theory to the field data of Rawlins and





Gardner (1963) proved successful, yielding fractional values of m less than unity in $t^m$. In a field
experimental study of horizontal water absorption into porous construction materials (fired-clay
and siliceous brick), El-Abd and Milczarek (2004) arrived at a formulation of diffusivity
coefficient again in the form $D(\theta,t) = E(\theta) \, t^m$. The application of this form to their experimental
data produced satisfactory results.
The study by Pachepsky et al. (2003) appears to be the first to propose a fractional model of
horizontal, unsaturated soil water flow in field soils. Motivated by the observations of Nielsen et
al. (1962) on the jerky movements of the infiltration front in field soils, that can be explained by
long recurrence time intervals in-between motions, Pachepsky et al. (2003) proposed a time-
fractional model of horizontal soil water flow in field soils. While the space component of their
model has integer derivatives, they proposed a fractional form for the diffusivity, and expressed
the Darcy water flux formulation in diffusive form with their proposed fractional diffusivity.
Pachepsky et al. (2003) showed that the cause for fractional diffusivity is the scaling of time in
the Boltzmann relationship not with the power of 0.5 (which corresponds to Brownian motion)
but with a power less than 0.5, an experimental observation that was already made by Guerrini
and Swartzendruber (1992). Pachepsky et al. (2003) supported their claim by various previous
experimental studies' results, and showed that their proposed time-fractional form of the
Richards equation with fractional diffusivity can explain experimental data. Meanwhile,
Gerolymatou et al. (2006) proposed a fractional integral form for the Richards equation in fractional
time but in integer horizontal space for unsaturated soil water flow in one horizontal dimension.
Comparing their model simulations against the field experimental data of El-Abd and Milczarek
(2004), they showed that their fractional Richards equation describes the evolution of soil water
content in time and space better than the corresponding integer Richards equation. Again





considering horizontal unsaturated soil water flow in fractional time but integer space, Sun et al.
(2013) utilized the concept of fractal ruler in time, due to Cushman et al. (2009), to define a
fractional derivative in time which they used to modify the integer time derivative in the
conventional Richards equation. By means of this fractional derivative definition they were able
to model the anomalous Boltzmann scaling in the wetting front movement and were able to
obtain good fits to water content experimental data. Sun et al. (2013) conjectured that the time-
dependent diffusivity $D(\theta,t) = E(\theta) \, t^m$ (for a fractional value of m) due to Guerrini and
Swartzendruber (1992) and El Abd and Milczarek (2004), in the conventional Richards equation
can be expressed essentially by representing the conventional integer derivative of the soil water
content with respect to time by a product of the fractional time derivative of the soil water
content and a fractional power of time.
The above-cited studies on the governing equations of soil water flow only treat time with
fractional dimension, while keeping space with integer dimension. Furthermore, these studies
address only one spatial dimension. Accordingly, our study in the following will attempt to
develop a fractional continuity equation and a fractional water flux (motion) equation for
unsaturated soil water flow in both fractional time and in multi-dimensional fractional space,
starting from the conventional mass conservation and Darcy's law. Due to the anisotropy in the
hydraulic conductivities of natural soils, the soil medium within which the soil water flow occurs
is essentially anisotropic. Accordingly, in this study the fractional dimensions in two horizontal
and one vertical directions will be considered different, resulting in multi-fractional space within
which the flow takes place. Toward the development of the fractional governing equations, first a
dimensionally-consistent continuity equation for soil water flow in multi-fractional, multi-
dimensional space and fractional time will be developed. For the motion equation of soil water



flow, or the equation of water flux within the soil matrix in multi-fractional multi-dimensional
space and fractional time, a dimensionally consistent equation will also be  developed. From the
combination of the fractional continuity and motion equations, the governing equation of
transient soil water flow in multi-fractional, multi-dimensional space and fractional time will be
obtained. It will be shown that this equation approaches the conventional Richards equation as
the fractional derivative powers approach integer values. Then by the introduction of the Brooks-
Corey constitutive relationships for soil water (Brooks and Corey, 1964) into the fractional
transient soil water flow equation, an explicit form of the equation will  be obtained in multi-
dimensional, multi-fractional space and fractional time. The governing fractional equation is then
specialized to the case of only vertical soil water flow and of only horizontal soil water flow in
fractional time-space.

DERIVATION OF THE CONTINUITY EQUATION FOR TRANSIENT SOIL WATER
FLOW IN MULTI-DIMENSIONAL FRACTIONAL SPACE AND FRACTIONAL TIME
The fractional Taylor series approximation of a function f(x) around x may be defined

according to the generalized Taylor series formula (Odibat and Shawagfeh, 2007;  Momani and
Odibat, 2008) as:
$$f(x + \Delta x) \cong \sum_{k=0}^{n} \frac{(x+\Delta x-x)^{k\beta}}{\Gamma(k\beta+1)} D_0^{k\beta} f(x), \ \ 0 < \beta \leq 1 \tag{1}$$
where $\Gamma(\cdot)$ is the gamma function, and $D_x^{k\beta} f(y)$, is a left-sided Caputo fractional derivative of
the function f (y), defined as (Odibat and Shawagfeh, 2007; Podlubny, 1999),
$$D_x^{k\beta} f(y) = \frac{1}{\Gamma(m-k\beta)} \int_x^y \frac{f^m(\xi)}{(y-\xi)^{k\beta+1-m}} d\xi, \qquad m\text{-}1 < \beta < m, \ m\epsilon N, \ y \geq x \ . \tag{2}$$





Specializing the integer $m =1$ reduces equation (2) to
$D_x^{k\beta} f(y) = \frac{1}{\Gamma(1-k\beta)} \int_x^y \frac{f^`(\xi)}{(y-\xi)^{k\beta}} d\xi,$          $0 < \beta < 1, \quad y \geq x$ .      (3)
Then to β-order
$D_x^{\beta} f(y) = \frac{1}{\Gamma(1-\beta)} \int_x^y \frac{f^`(\xi)}{(y-\xi)^{\beta}} d\xi$          $0 < \beta < 1, \quad y \geq x$ .      (4)
Specializing the fractional Taylor series expansion to β-order (k = 0, 1 in equation (1)), one
obtains from the generalized Taylor series formula:
$f(x + \Delta x) = f(x) + \frac{(\Delta x)^{\beta}}{\Gamma(\beta+1)} D_0^{\beta} f(x), \quad 0 < \beta \leq 1$      (5)
to β-order.

Within the above framework one can express the net mass outflow rate from the control

volume in Figure 1 as
$\left[\rho q_{x_1}(x_1 + \Delta x_1, x_2, x_3; t) - \rho q_{x_1}(x_1, x_2, x_3; t)\right]\Delta x_2 \Delta x_3 + \left[\rho q_{x_2}(x_1, x_2 + \Delta x_2, x_3; t) - \right.$
$\rho q_{x_2}(x_1, x_2, x_3; t)]\Delta x_1 \Delta x_3 + \left[\rho q_{x_3}(x_1, x_2, x_3 + \Delta x_3; t) - \rho q_{x_3}(x_1, x_2, x_3; t)\right]\Delta x_1 \Delta x_2$      (6)
Then by introducing equation (5) into equation (6), and expressing the Caputo derivative
$D_0^{\beta} f(x)$ by $\frac{\partial^{\beta} f(x)}{(\partial x)^{\beta}}$ for convenience, the net mass flux from the soil control volume in Figure 1
may be expressed to β-order in fractional space as,
$= \frac{(\Delta x_1)^{\beta_1}}{\Gamma(\beta_1+1)} \left(\frac{\partial}{\partial x_1}\right)^{\beta_1} \left(\rho q_{x_1}(x_1, x_2, x_3; t)\right) \Delta x_2 \Delta x_3 + \frac{(\Delta x_2)^{\beta_2}}{\Gamma(\beta_2+1)} \left(\frac{\partial}{\partial x_2}\right)^{\beta_2} \left(\rho q_{x_2}(x_1, x_2, x_3; t)\right) \Delta x_1 \Delta x_3$





$\quad + \frac{(\Delta x_3)^{\beta_3}}{\Gamma(\beta_3+1)} \left( \frac{\partial}{\partial x_3} \right)^{\beta_3} \left( \rho q_{x_3}(x_1, x_2, x_3; t) \right) \Delta x_1 \Delta x_2$ $\qquad$ (7)
$\quad$ where different fractional powers are considered in the three Cartesian directions in space due to
$\quad$ the general anisotropy in the soil permeabilities and in the resulting flows in the soil media. It
$\quad$ also follows from equation (5) with $f(x_i) = x_i$ that to $\beta$-order,

$\quad \Delta x_i = \frac{(\Delta x_i)^{\beta_i}}{\Gamma(\beta_i+1)} \frac{\partial^{\beta_i} x_i}{(\partial x_i)^{\beta_i}}$ $\qquad$ i=1,2,3 $\qquad$ (8)
$\quad$ With respect to the Caputo derivative;
$\quad \frac{\partial^{\beta_i} x_i}{(\partial x_i)^{\beta_i}} = \frac{x_i^{1-\beta_i}}{\Gamma(2-\beta_i)}$ , $\qquad$ i=1,2,3 $\qquad$ (9)
$\quad$ Hence, combining equations (8) and (9) yields,
$\quad (\Delta x_i)^{\beta_i} = \frac{\Gamma(\beta_i+1)\Gamma(2-\beta_i)}{x_i^{1-\beta_i}} (\Delta x_i),$ $\qquad$ i=1,2,3 $\qquad$ (10)
$\quad$ with respect to $\beta_i$-order fractional space in the i-th direction, i=1,2,3.
$\quad$ Introducing equation (10) into equation (7) yields for the net mass outflow rate
$\quad = \frac{\Gamma(2-\beta_1)}{x_1^{1-\beta_1}} \left( \frac{\partial}{\partial x_1} \right)^{\beta_1} \left( \rho q_{x_1}(\bar{x}; t) \right) \Delta x_1 \Delta x_2 \Delta x_3 + \frac{\Gamma(2-\beta_2)}{x_2^{1-\beta_2}} \left( \frac{\partial}{\partial x_2} \right)^{\beta_2} \left( \rho q_{x_2}(\bar{x}; t) \right) \Delta x_1 \Delta x_2 \Delta x_3$
$\quad + \frac{\Gamma(2-\beta_3)}{x_3^{1-\beta_3}} \left( \frac{\partial}{\partial x_3} \right)^{\beta_3} \left( \rho q_{x_3}(\bar{x}; t) \right) \Delta x_1 \Delta x_2 \Delta x_3$ , $\qquad \bar{x} = (x_1, x_2, x_3)$ $\qquad$ (11)
$\quad$ to $\beta$-order, reflecting multi-fractional scaling in the anisotropic soil medium.
$\quad\quad$ Denoting the volumetric water content by $\theta(\bar{x},t)$, the water volume $V_w$ within the control
$\quad$ volume in Figure 1 may be expressed as
$\quad V_w = \theta \, \Delta x_1 \Delta x_2 \Delta x_3$ $\qquad$ . $\qquad$ (12)
$\quad$ Hence, the time rate of change of mass within the control volume in Figure 1 is





$\quad \lim_{\Delta t \to 0} \frac{\rho(\bar{x},t+\Delta t)\theta(\bar{x},t+\Delta t) - \rho(\bar{x},t)\theta(\bar{x},t)}{\Delta t} \Delta x_1 \Delta x_2 \Delta x_3$ . (13)
Introducing equation (5) with x replaced by t, into equation (13) yields the time rate of change of
mass within the control volume with respect $\alpha$-fractional time increments:
$\quad \lim_{\Delta t \to 0} \frac{(\Delta t)^\alpha}{\Delta t \, \Gamma(\alpha+1)} \left(\frac{\partial}{\partial t}\right)^\alpha \rho(\bar{x},t)\theta(\bar{x},t)$ . (14)
to $\alpha$-order. With respect to the Caputo derivative:
$\quad \frac{\partial^\alpha t}{(\partial t)^\alpha} = \frac{t^{1-\alpha}}{\Gamma(2-\alpha)}$ (15)
which when combined with equation (5) (with x replaced by t) yields
$\quad (\Delta t)^\alpha = \frac{\Gamma(\alpha+1)\Gamma(2-\alpha)}{t^{1-\alpha}} (\Delta t)$ . (16)
to $\alpha$-order. Introducing equation (16) into equation (14) yields for the time rate of change of
mass within the control volume in Figure 1 with respect to $\alpha$-order fractional time increments:
$\quad \frac{\Gamma(2-\alpha)}{t^{1-\alpha}} \frac{\partial^\alpha \rho(\bar{x},t)\theta(\bar{x},t)}{(\partial t)^\alpha} \Delta x_1 \Delta x_2 \Delta x_3$ . (17)
$\quad$ Since the time rate of change of mass within the control volume of Figure 1 is inversely
related to the net flux through the control volume, equations (11) and (17) can be combined to
yield
$\quad \frac{\Gamma(2-\alpha)}{t^{1-\alpha}} \frac{\partial^\alpha \rho(\bar{x},t)\theta(\bar{x},t)}{(\partial t)^\alpha} = - \left[ \frac{\Gamma(2-\beta_1)}{x_1^{1-\beta_1}} \left(\frac{\partial}{\partial x_1}\right)^{\beta_1} \left(\rho q_{x_1}(\bar{x};t)\right) + \frac{\Gamma(2-\beta_2)}{x_2^{1-\beta_2}} \left(\frac{\partial}{\partial x_2}\right)^{\beta_2} \left(\rho q_{x_2}(\bar{x};t)\right) + \right.$
$\quad\quad\quad\quad\quad\quad \left. \frac{\Gamma(2-\beta_3)}{x_3^{1-\beta_3}} \left(\frac{\partial}{\partial x_3}\right)^{\beta_3} \left(\rho q_{x_3}(\bar{x};t)\right) \right]$ ,
$\quad \frac{\Gamma(2-\alpha)}{t^{1-\alpha}} \frac{\partial^\alpha \rho(\bar{x},t)\theta(\bar{x},t)}{(\partial t)^\alpha} = - \sum_{i=1}^{3} \frac{\Gamma(2-\beta_i)}{x_i^{1-\beta_i}} \left(\frac{\partial}{\partial x_i}\right)^{\beta_i} \left(\rho(\bar{x};t)q_{x_i}(\bar{x};t)\right)$ (18)
as the fractional continuity equation of transient soil water flow in multi-fractional space of a
generally anisotropic soil medium in fractional time.




If one further assumes an incompressible soil medium with constant density, then the
fractional soil water flow continuity equation (18) simplifies further to
$\frac{\Gamma(2-\alpha)}{t^{1-\alpha}}\frac{\partial^{\alpha}\theta(\bar{x},t)}{(\partial t)^{\alpha}} = -\sum_{i=1}^{3}\frac{\Gamma(2-\beta_i)}{x_i^{1-\beta_i}}\left(\frac{\partial}{\partial x_i}\right)^{\beta_i}\left(q_{x_i}(\bar{x};t)\right)$ , $0<\alpha, \beta_1, \beta_2, \beta_3 <1$ ; $\bar{x}=(x_1, x_2, x_3)$.     (19)
In the following, Equation (19) will be used as the fractional continuity equation for soil water
flow for further study.
Performing a dimensional analysis of Equation (19), one obtains
$\frac{1}{T^{1-\alpha}}\cdot\frac{1}{T^{\alpha}} = \frac{1}{L^{1-\beta_i}}\frac{1}{L^{\beta_i}}\frac{L}{T} = \frac{1}{T}$     (20)
where L denotes length and T denotes time. Hence, Equation (20) shows the dimensional
consistency of the left hand and right hand sides of the continuity Equation (19) for transient soil
water flow in multi-fractional space and fractional time.
Podlubny (1999) has shown that for $n$-1$< \alpha, \beta_i < n$ where n is any positive integer, as
$\alpha$ and $\beta_i \rightarrow$ n, the Caputo fractional derivative of a function f(y) to order $\alpha$ or $\beta_i$ (i = 1, 2, 3)
becomes the conventional n-th derivative of the function f(y). Therefore, specializing Podlubny's
(1999) result to n = 1, for $\alpha$ and $\beta_i \rightarrow$1  (i = 1, 2, 3), the continuity equation (19) reduces to
$\frac{\partial\theta(\bar{x},t)}{\partial t} = -\sum_{i=1}^{3}\frac{\partial}{\partial x_i}\left(q_{x_i}(\bar{x};t)\right)$     (21)
which is the conventional continuity equation for soil water flow.
AN EQUATION FOR SOIL WATER FLUX (SPECIFIC DISCHARGE) IN FRACTIONAL
TIME-SPACE
The experiments of Darcy (1856) showed that the specific discharge $q_i$ is directly
proportional to the change in hydraulic head, $\Delta h = h(x_i + \Delta x) - h(x_i)$, i=1,2,3, and is inversely
proportional to the spatial displacement in any direction i, $\Delta x_i = (x_i + \Delta x_i) - x_i$, i= 1,2,3 (Freeze
and Cherry, 1979). Hence, one can express the Darcy law in integer time-space as





$q_{x_i}\Delta x_i = -K_i \Delta h_i$    , $i = 1,2,3$ .                                    (22)
where $K_i = K_i(\bar{x})$ denotes the hydraulic conductivity in the i-th spatial direction (i=1,2,3), and the
negative sign on the right-hand-side (RHS) of equation (22) is due to soil water flow being in the
direction of decreasing hydraulic head.
In equation (22), using the fractional Taylor series expansion (5) to $\beta_i$-order (i= 1,2,3) yields:
$\Delta h_i = \frac{(\Delta x_i)^{\beta_i}}{\Gamma(\beta_i+1)} \frac{\partial^{\beta_i} h}{(\partial x_i)^{\beta_i}}$    , $i = 1, 2, 3$                                    (23)
where the notation is the same as above. Combining equations (8), (10) and (23) with equation
(22) yields,
$q_i \left[ \frac{x_i^{1-\beta_i}}{\Gamma(2-\beta_i)} + \frac{O((\Delta x_i)^{\beta_i})}{(\Delta x_i)^{\beta_i}} \Gamma(\beta_i + 1) \right] = -K_i \left[ \frac{\partial^{\beta_i} h}{(\partial x_i)^{\beta_i}} + \frac{O((\Delta x_i)^{\beta_i})}{(\Delta x_i)^{\beta_i}} \Gamma(\beta_i + 1) \right]$ , i=1,2,3 .          (24)

Taking the limit as $\Delta x_i$ goes to zero (i= 1,2,3), one obtains from equation (24),
$q_i(\bar{x}, t) = -K_i(\bar{x}) \frac{\Gamma(2-\beta_i)}{x_i^{1-\beta_i}} \frac{\partial^{\beta_i} h}{(\partial x_i)^{\beta_i}}$          i = 1,2,3                                    (25)
as the equation of water flux through anisotropic soil media in multi-fractional multi-dimensional
space.

Performing a dimensional analysis on equation (25), one obtains:

$[q_i(\bar{x}, t)] = {L}/{T}$      and    $\left[ K_i(\bar{x}) \frac{\Gamma(2-\beta_i)}{x_i^{1-\beta_i}} \frac{\partial^{\beta_i} h}{(\partial x_i)^{\beta_i}} \right] = \frac{L}{T} \frac{L}{L^{1-\beta_i} L^{\beta_i}} = \frac{L}{T}$                    (26)
which establishes the dimensional consistency of equation (25) as the fractional equation for soil
water flux. Furthermore, it is well-known that for unsaturated soil water flow, the hydraulic
conductivity is function of the volumetric soil water content θ and of spatial location (Freeze and
Cherry, 1979). In fact, $K_i$ may be expressed in terms of the saturated hydraulic conductivity $K_s$
and the relative hydraulic conductivity $K_r(\theta)$ as





$K_i(\overline{x}, \theta) = K_{s,i}(\overline{x})K_r(\theta)$           .                                (27)
Hence, the equation of soil water flux (specific discharge) in multi-dimensional, multi-fractional
anisotropic soil space may be expressed as
$q_i(\bar{x}, t) = -K_i(\overline{x}, \theta) \dfrac{\Gamma(2-\beta_i)}{x_i^{1-\beta_i}} \dfrac{\partial^{\beta_i} h(\overline{x}, t)}{(\partial x_i)^{\beta_i}}$         ,   i= 1,2,3 .                     (28)
Equation (28) is dimensionally consistent in that both the LHS and RHS of the equation have the
unit L/T.

As noted above, Podlubny (1999) has shown that for $n$-1$< \beta_i < n$ (i = 1, 2, 3) where n is any

positive integer, as $\beta_i \rightarrow n$, the Caputo fractional derivative of a function f(y) to order $\beta_i$ (i = 1, 2,
3) becomes the conventional n-th derivative of the function f(y). Therefore, specializing
Podlubny's (1999) result to n = 1, for $\beta_i \rightarrow 1$ (i = 1, 2, 3), the fractional soil water flux equation
(28) becomes
$q_i(\bar{x}, t) = -K_i(\overline{x}, \theta) \dfrac{\partial h(\overline{x}, t)}{\partial x_i}$            ,     i= 1,2,3 .                   (29)
which is the conventional Darcy's equation for soil water flux. As such the derived fractional soil
water flux Equation (28) is consistent with the conventional Darcy's equation for the integer
power case.
GOVERNING EQUATION OF TRANSIENT SOIL WATER FLOW IN MULTI-
DIMENSIONAL FRACTIONAL SOIL SPACE AND FRACTIONAL TIME
Combining the fractional continuity equation (19) with the fractional soil water flux equation

(28) yields,
$\dfrac{\Gamma(2-\alpha)}{t^{1-\alpha}} \dfrac{\partial^{\alpha} \theta(\bar{x}, t)}{(\partial t)^{\alpha}} = \sum_{i=1}^{3} \dfrac{\Gamma(2-\beta_i)}{x_i^{1-\beta_i}} \left(\dfrac{\partial}{\partial x_i}\right)^{\beta_i} \left( K_i(\overline{x}, \theta) \dfrac{\Gamma(2-\beta_i)}{x_i^{1-\beta_i}} \dfrac{\partial^{\beta_i} h(\overline{x}, t)}{(\partial x_i)^{\beta_i}} \right)$ for 0< $\alpha$, $\beta_1$, $\beta_2$, $\beta_3$ <1 ;
$\bar{x} = (x_1, x_2, x_3)$.                                                             (30)




Since $K_i(\bar{x}, \theta) = K_{s,i}(\bar{x}) K_r(\theta)$, one obtains
$\frac{\Gamma(2-\alpha)}{t^{1-\alpha}} \frac{\partial^{\alpha}\theta(\bar{x},t)}{(\partial t)^{\alpha}} = \sum_{i=1}^{3} \frac{\Gamma(2-\beta_i)}{x_i^{1-\beta_i}} \left(\frac{\partial}{\partial x_i}\right)^{\beta_i} \left(K_{s,i}(\bar{x}) K_r(\theta) \frac{\Gamma(2-\beta_i)}{x_i^{1-\beta_i}} \frac{\partial^{\beta_i}h(\bar{x},t)}{(\partial x_i)^{\beta_i}}\right)$ for $0< \alpha, \beta_1, \beta_2, \beta_3 <1$;
$$\bar{x} = (x_1, x_2, x_3) \qquad (31)$$
as the governing equation of transient soil water flow in anisotropic multi-dimensional fractional
soil media and fractional time.
Meanwhile, the soil hydraulic head h is related to the elevation head $x_3$ and soil capillary
pressure head ψ by
$h = \psi(\theta) + x_3$ (32)
Substituting Equation (32) into Equation (31) results in
$\frac{\Gamma(2-\alpha)}{t^{1-\alpha}} \frac{\partial^{\alpha}\theta(\bar{x},t)}{(\partial t)^{\alpha}} = \sum_{i=1}^{3} \frac{\Gamma(2-\beta_i)}{x_i^{1-\beta_i}} \left(\frac{\partial}{\partial x_i}\right)^{\beta_i} \left(K_{s,i}(\bar{x}) K_r(\theta) \frac{\Gamma(2-\beta_i)}{x_i^{1-\beta_i}} \frac{\partial^{\beta_i}}{(\partial x_i)^{\beta_i}} (\psi(\theta) + x_3)\right)$ . (33)

With respect to the Caputo derivative:
$\frac{\partial^{\beta_3}x_3}{(\partial x_3)^{\beta_3}} = \frac{x_3^{1-\beta_3}}{\Gamma(2-\beta_3)}$ . (34)
Opening equation (33) further and introducing equation (34) yields
$\frac{\Gamma(2-\alpha)}{t^{1-\alpha}} \frac{\partial^{\alpha}\theta(\bar{x},t)}{(\partial t)^{\alpha}} = \sum_{i=1}^{3} \frac{\Gamma(2-\beta_i)}{x_i^{1-\beta_i}} \left(\frac{\partial}{\partial x_i}\right)^{\beta_i} \left(K_{s,i}(\bar{x}) K_r(\theta) \frac{\Gamma(2-\beta_i)}{x_i^{1-\beta_i}} \frac{\partial^{\beta_i}\psi(\theta)}{(\partial x_i)^{\beta_i}}\right)$
$+ \frac{\Gamma(2-\beta_3)}{x_3^{1-\beta_3}} \left(\frac{\partial}{\partial x_3}\right)^{\beta_3} \left(K_{s,3}(\bar{x}) K_r(\theta)\right)$; $0< \alpha, \beta_1, \beta_2, \beta_3 <1$ ; $\bar{x} = (x_1, x_2, x_3)$ (35)
as the governing equation of transient soil water flow in anisotropic multi-dimensional fractional
media and fractional time. This governing equation may also be written as
$\frac{\partial^{\alpha}\theta(\bar{x},t)}{(\partial t)^{\alpha}} = \sum_{i=1}^{3} \frac{1}{\Gamma(2-\alpha)} \frac{(\Gamma(2-\beta_i))^2}{x_i^{1-\beta_i}} \left(\frac{\partial}{\partial x_i}\right)^{\beta_i} \left(K_{s,i}(\bar{x}) K_r(\theta) \frac{t^{1-\alpha}}{x_i^{1-\beta_i}} \frac{\partial^{\beta_i}\psi(\theta)}{(\partial x_i)^{\beta_i}}\right)$
$+ \frac{1}{\Gamma(2-\alpha)} \frac{\Gamma(2-\beta_3)}{x_3^{1-\beta_3}} \left(\frac{\partial}{\partial x_3}\right)^{\beta_3} \left(t^{1-\alpha} K_{s,3}(\bar{x}) K_r(\theta)\right)$ ; $0< \alpha, \beta_1, \beta_2, \beta_3 <1$ ; $\bar{x} = (x_1, x_2, x_3)$ .(36)





As noted above, Podlubny (1999) has shown that for $n$-1$< \alpha, \beta_i < n$  (i=1,2,3) where n is any
positive integer, as  α and $\beta_i \rightarrow$ n, the Caputo fractional derivative of a function f(y) to order α or
$\beta_i$ (i = 1, 2, 3) becomes the conventional n-th derivative of the function f(y). Therefore,
specializing Podlubny's (1999) result to n = 1, for α and $\beta_i \rightarrow$1  (i = 1, 2, 3), the fractional
governing equation (33) of soil water flow becomes
$$\frac{\partial \theta(\bar{x},t)}{\partial t} = \sum_{i=1}^{3} \frac{\partial}{\partial x_i} \left( K_{s,i}(\bar{x}) K_r(\theta) \frac{\partial}{\partial x_i} (\psi(\theta) + x_3) \right) \qquad (37)$$
which is the conventional Richards equation for transient soil water flow.
With respect to dimensional consistency, one may note that both sides of the fractional
governing equation (33) or equation (35) for transient soil water flow have the unit 1/T.
Meanwhile, both sides of equation (36) have the unit $1/T^{\alpha}$. Hence, these fractional equations are
dimensionally consistent.

FRACTIONAL GOVERNING EQUATION OF TRANSIENT SOIL WATER FLOW IN THE
VERTICAL DIRECTION

In the case of vertical transient unsaturated flow for the infiltration process in soils in
fractional time-space, Equation (35) reduces further to
$$\frac{\Gamma(2-\alpha)}{t^{1-\alpha}} \frac{\partial^{\alpha} \theta(\bar{x},t)}{(\partial t)^{\alpha}} = \frac{\Gamma(2-\beta_3)}{x_3^{1-\beta_3}} \left( \frac{\partial}{\partial x_3} \right)^{\beta_3} \left( K_{s,3}(\bar{x}) K_r(\theta) \frac{\Gamma(2-\beta_3)}{x_3^{1-\beta_3}} \frac{\partial^{\beta_3} \psi(\theta)}{(\partial x_3)^{\beta_3}} \right) +$$
$$+ \frac{\Gamma(2-\beta_3)}{x_3^{1-\beta_3}} \left( \frac{\partial}{\partial x_3} \right)^{\beta_3} \left( K_{s,3}(\bar{x}) K_r(\theta) \right) \qquad ; 0 < \alpha, \beta_3 < 1 ; \bar{x} = (x_1, x_2, x_3) \qquad (38)$$
as the governing equation. This governing equation for vertical transient soil water flow in
fractional time-space can also be expressed as;
$$\frac{\partial^{\alpha} \theta(\bar{x},t)}{(\partial t)^{\alpha}} = \frac{\Gamma(2-\beta_3)}{x_3^{1-\beta_3}} \left( \frac{\partial}{\partial x_3} \right)^{\beta_3} \left( K_{s,3}(\bar{x}) K_r(\theta) \frac{\Gamma(2-\beta_3)}{\Gamma(2-\alpha)} \frac{t^{1-\alpha}}{x_3^{1-\beta_3}} \frac{\partial^{\beta_3} \psi(\theta)}{(\partial x_3)^{\beta_3}} \right) +$$



$$+ \frac{1}{\Gamma(2-\alpha)} \frac{\Gamma(2-\beta_3)}{x_3^{1-\beta_3}} \left(\frac{\partial}{\partial x_3}\right)^{\beta_3} \left(t^{1-\alpha} K_{s,3}(\overline{x}) K_r(\theta)\right) \quad ; 0 < \alpha, \beta_3 < 1 \; ; \overline{x} = (x_1, x_2, x_3). \qquad (39)$$


As in the integer case of Richards equation (37), equations (35), (36), (38) and (39) have both
the hydraulic conductivity K and the capillary pressure head ψ as functions of the soil
volumetric water content θ. As such, characteristic soil water relationships, such as those given
by Brooks and Corey (1964), may be utilized to obtain an explicit form of the governing
equation of transient, unsaturated soil water flow in fractional time-space, as explained in the
following.

SOIL WATER CONTENT-BASED EXPLICIT FORM OF THE GOVERNING EQUATION
OF TRANSIENT SOIL WATER FLOW IN FRACTIONAL TIME-SPACE

One can utilize the Brooks-Corey (1964) formula for the soil characteristic relationship

between the capillary soil water pressure head ψ and the soil water content θ as follows:
$\psi(\theta) = \psi_b \theta_e^{1/\lambda} (\theta - \theta_r)^{-1/\lambda}$ \hfill (40)
where $\psi_b$ is the air entry pressure head (bubbling pressure), $\theta_e = (\theta_s - \theta_r)$ is the effective porosity,
$\theta_s$ is the saturation volumetric soil water content, $\theta_r$ is the residual water content, and $\lambda$ is the
pore size distribution index. Therefore, the $\beta_i$-order Caputo fractional derivative of the capillary
pressure head ψ with respect to $x_i$ in the interval $(0, x_i)$ may be expressed in terms of the Brooks-
Corey relationship (40) as (Podlubny, 1999; Odibat and Shawagfeh, 2007)
$$\frac{\partial^{\beta_i} \psi(\theta)}{(\partial x_i)^{\beta_i}} = \frac{\psi_b \theta_e^{1/\lambda}}{\Gamma(1-\beta_i)} \int_0^{x_i} \left(\frac{\partial}{\partial \xi_i} (\theta - \theta_r)^{-1/\lambda}\right) (x_i - \xi_i)^{-\beta_i} d\xi_i = \psi_b \theta_e^{1/\lambda} \frac{\partial^{\beta_i}(\theta - \theta_r)^{-1/\lambda}}{(\partial x_i)^{\beta_i}}. \qquad (41)$$
Under the Brooks-Corey (1964) relationship between the hydraulic conductivity and the
volumetric soil water content, the relative hydraulic conductivity $K_r(\theta)$ is expressed as

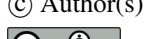



$\quad K_r(\theta) = \theta_e^{-3-2\lambda}(\theta - \theta_r)^{3+2\lambda}$ $\qquad$ , (42)
and using expression (42) within $K_i(\overline{x}, \theta) = K_{s,i}(\overline{x})K_r(\theta)$ , the $\beta_i$-order fractional Caputo
derivative of $K_i(\overline{x}, \theta)$ with respect to $x_i$ in the interval $(0, x_i)$ may be expressed as
$\quad \dfrac{\partial^{\beta_i} K_{s,i}(\overline{x}) K_r(\theta)}{(\partial x_i)^{\beta_i}} = \theta_e^{-3-2\lambda} \dfrac{\partial^{\beta_i}\left(K_{s,i}(\overline{x})(\theta-\theta_r)^{3+2\lambda}\right)}{(\partial x_i)^{\beta_i}}$ $\quad$ , i = 1,2,3 . (43)
Substituting equations (41) and (43) into equation (35) results in an explicit form of the
governing equation of transient soil water flow in anisotropic multi-dimensional fractional soil
space and fractional time in terms of the volumetric water content $\theta$ as
$\quad \dfrac{\Gamma(2-\alpha)}{t^{1-\alpha}}\dfrac{\partial^{\alpha}\theta(\bar{x},t)}{(\partial t)^{\alpha}} = \sum_{i=1}^{3}\psi_b\theta_e^{-3-2\lambda+1/\lambda}\dfrac{(\Gamma(2-\beta_i))^2}{x_i^{1-\beta_i}}\left(\dfrac{\partial}{\partial x_i}\right)^{\beta_i}\left(K_{s,i}(\overline{x})\dfrac{(\theta-\theta_r)^{3+2\lambda}}{x_i^{1-\beta_i}}\dfrac{\partial^{\beta_i}(\theta-\theta_r)^{-1/\lambda}}{(\partial x_i)^{\beta_i}}\right)$
$\quad\quad\quad + \theta_e^{-3-2\lambda}\dfrac{\Gamma(2-\beta_3)}{x_3^{1-\beta_3}}\left(\dfrac{\partial}{\partial x_3}\right)^{\beta_3}\left(K_{s,3}(\overline{x})(\theta-\theta_r)^{3+2\lambda}\right)$ $\quad$ ; 0< $\alpha,\beta_1, \beta_2, \beta_3$ <1 (44)
in terms of the Brooks-Corey soil water characteristics relationships. This governing equation can
also be expressed as
$\quad \dfrac{\partial^{\alpha}\theta(\bar{x},t)}{(\partial t)^{\alpha}} = \sum_{i=1}^{3}\psi_b\theta_e^{-3-2\lambda+1/\lambda}\dfrac{(\Gamma(2-\beta_i))^2}{\Gamma(2-\alpha)x_i^{1-\beta_i}}\left(\dfrac{\partial}{\partial x_i}\right)^{\beta_i}\left(K_{s,i}(\overline{x})(\theta-\theta_r)^{3+2\lambda}\dfrac{t^{1-\alpha}}{x_i^{1-\beta_i}}\dfrac{\partial^{\beta_i}(\theta-\theta_r)^{-1/\lambda}}{(\partial x_i)^{\beta_i}}\right)$
$\quad\quad\quad + \theta_e^{-3-2\lambda}\dfrac{\Gamma(2-\beta_3)}{\Gamma(2-\alpha)x_3^{1-\beta_3}}\left(\dfrac{\partial}{\partial x_3}\right)^{\beta_3}\left(t^{1-\alpha}K_{s,3}(\overline{x})(\theta-\theta_r)^{3+2\lambda}\right)$ $\quad$ ; 0< $\alpha,\beta_1, \beta_2, \beta_3$ <1. (45)

Upon dimensional analysis of equation (44) one can see that it is dimensionally consistent since
both of its sides have the unit of 1/T where T denotes time. Meanwhile, equation (45) is also
dimensionally consistent with both sides of the equation having the unit $1/T^{\alpha}$.
$\quad\quad$ Specializing equation (45) to only the vertical direction, the governing equation of
transient soil water flow in the vertical direction in fractional space-time may be expressed as,
$\quad \dfrac{\partial^{\alpha}\theta(\bar{x},t)}{(\partial t)^{\alpha}} = \psi_b\theta_e^{-3-2\lambda+1/\lambda}\dfrac{(\Gamma(2-\beta_3))^2}{\Gamma(2-\alpha)x_3^{1-\beta_3}}\left(\dfrac{\partial}{\partial x_3}\right)^{\beta_3}\left(K_{s,3}(\overline{x})(\theta-\theta_r)^{3+2\lambda}\dfrac{t^{1-\alpha}}{x_3^{1-\beta_3}}\dfrac{\partial^{\beta_3}(\theta-\theta_r)^{-1/\lambda}}{(\partial x_3)^{\beta_3}}\right) +$



$$+ \theta_e^{-3-2\lambda} \frac{\Gamma(2-\beta_3)}{\Gamma(2-\alpha)x_3^{1-\beta_3}} \left(\frac{\partial}{\partial x_3}\right)^{\beta_3} \left(t^{1-\alpha}K_{s,3}(\overline{x})(\theta-\theta_r)^{3+2\lambda}\right) \quad ; 0< \alpha, \beta_3 <1 \; . \quad (46)$$
Upon dimensional analysis of equation (46) one can find that both sides of this equation have
the unit of $1/T^{\alpha}$ where T denotes time. Hence, the fractional equation of vertical transient soil
water flow, in its explicit form, is dimensionally consistent.
Finally, specializing equation (45) to only the horizontal directions, the governing equation
of transient soil water flow in the horizontal directions in fractional space-time may be expressed
as,
$$\frac{\partial^{\alpha}\theta(\bar{x},t)}{(\partial t)^{\alpha}} = \sum_{i=1}^{2} \psi_b \theta_e^{-3-2\lambda+1/\lambda} \frac{(\Gamma(2-\beta_i))^2}{\Gamma(2-\alpha)x_i^{1-\beta_i}} \left(\frac{\partial}{\partial x_i}\right)^{\beta_i} \left(K_{s,i}(\overline{x})(\theta-\theta_r)^{3+2\lambda} \frac{t^{1-\alpha}}{x_i^{1-\beta_i}} \frac{\partial^{\beta_i}(\theta-\theta_r)^{-1/\lambda}}{(\partial x_i)^{\beta_i}}\right)$$
$$; 0< \alpha, \beta_1, \beta_2 <1 \; . \quad (47)$$
Upon dimensional analysis of equation (47) one can find that both sides of this equation have the
unit of $1/T^{\alpha}$ where T denotes time. Hence, the fractional equation of horizontal transient soil
water flow, in its explicit form, is dimensionally consistent.

DISCUSSION AND CONCLUSION
The governing equations that were developed in this study are for the fractional time
dimension and for multi-dimensional fractional space that represents the fractal spatial structure
of a soil field. If one were to simplify the developed theory above to only fractional time but
integer-space soil fields, then the developed equations would simplify substantially. The
governing equation (36) of transient soil water flow in anisotropic multi-dimensional fractional
soil media in fractional time would simplify to (with $\beta_i = 1$, $i = 1,2,3$):
$$\frac{\partial^{\alpha}\theta(\bar{x},t)}{(\partial t)^{\alpha}} = \sum_{i=1}^{3} \frac{1}{\Gamma(2-\alpha)} \frac{\partial}{\partial x_i} \left(K_{s,i}(\overline{x})K_r(\theta)t^{1-\alpha} \frac{\partial\psi(\theta)}{\partial x_i}\right)$$





$$+ \frac{1}{\Gamma(2-\alpha)} \frac{\partial}{\partial x_3}\left(t^{1-\alpha} K_{s,3}(\overline{x}) K_r(\theta)\right) \quad ; 0< \alpha <1; \bar{x} = (x_1, x_2, x_3) \quad (48)$$

for the governing equation of transient soil water flow in integer multi-dimensional soil media
and in fractional time. In terms of the Brooks-Corey soil characteristic relationships, the explicit
governing equation of transient soil water flow in integer multi-dimensional soil space and in
fractional time is obtained from the simplification of equation (45) as (with $\beta_i = 1$, i =1,2,3):
$$\frac{\partial^{\alpha}\theta(\bar{x},t)}{(\partial t)^{\alpha}} = \sum_{i=1}^{3} -\frac{1}{\lambda} \psi_b \theta_e^{-3-2\lambda+1/\lambda} \frac{1}{\Gamma(2-\alpha)} \frac{\partial}{\partial x_i}\left(t^{1-\alpha} K_{s,i}(\overline{x})(\theta - \theta_r)^{2-1/\lambda+2\lambda} \frac{\partial\theta}{\partial x_i}\right)$$
$$+ \theta_e^{-3-2\lambda} \frac{1}{\Gamma(2-\alpha)} \frac{\partial}{\partial x_3}\left(t^{1-\alpha} K_{s,3}(\overline{x})(\theta - \theta_r)^{3+2\lambda}\right) \quad ; 0< \alpha <1 ; \bar{x} = (x_1, x_2, x_3). \qquad (49)$$

As mentioned before, Guerrini and Swartzendruber (1992) and El Abd and Milczarek (2004),
in their explanation of the anomalous behavior of the diffusivity coefficient in their experiments,
have proposed that the diffusivity coefficient in the diffusion-based formulation of the Richards
equation of soil water flow must depend not only on the water content but also on time. Hence,
they formulated this diffusivity coefficient D as D = D($\theta$,t) = E($\theta$) $t^m$ where E is a function of
water content $\theta$ while m is a power value. This formulation proved to be successful in modeling
various experimental data on horizontal soil water flow. If one were to formulate the diffusivity
$D_i(\theta,t)$ in the explicit governing equation (49) of transient soil water flow in fractional time and
in anisotropic multi-dimensional integer soil space as
$$D_i(\theta, t) = K_{s,i}(\overline{x})(\theta - \theta_r)^{2-1/\lambda+2\lambda} t^{1-\alpha} \qquad , \quad i = 1,2,3, \qquad (50)$$
this diffusivity coefficient is in the same form as the diffusivity coefficient D($\theta$,t) = E($\theta$) $t^m$ that
was formulated by Guerrini and Swartzendruber (1992) and El Abd and Milczarek (2004) based
on experimental observations. As such, within the framework of Brooks-Corey soil water
relationships, the  explicit governing equations that were developed in this study for the transient





soil water flow in multi-dimensional fractional soil media and fractional time, when simplified to
integer soil space, are consistent with the experimental observations of Guerrini and
Swartzendruber (1992) and El Abd and Milczarek (2004) when their power value m = 1-$\alpha$ .

Sun et al. (2013) conjectured that the time-dependent diffusivity D($\theta$,t) = E($\theta$) t$^m$ (for a

fractional value of m) due to Guerrini and Swartzendruber (1992) and El Abd and Milczarek
(2004), in the conventional Richards equation can be expressed essentially by representing the
conventional integer derivative of the soil water content with respect to time by a product of the
fractional time derivative of the soil water content and a fractional power of time (Sun et al.
2013, Eqn. (12)), that is, $\frac{\partial \theta(\bar{x},t)}{\partial t} = \frac{C}{t^{1-\alpha}} \frac{\partial^{\alpha} \theta(\bar{x},t)}{(\partial t)^{\alpha}}$ where C denotes a constant.  In order to examine
the conjecture of Sun et al. (2013), one can re-write the explicit governing equation (49) for soil
water flow in integer space but fractional time in equivalent form as
$\frac{\Gamma(2-\alpha)}{t^{1-\alpha}} \frac{\partial^{\alpha} \theta(\bar{x},t)}{(\partial t)^{\alpha}} = \sum_{i=1}^{3} -\frac{1}{\lambda} \psi_b \theta_e^{-3-2\lambda+\frac{1}{\lambda}} \frac{\partial}{\partial x_i} \left( K_{s,i}(\bar{x})(\theta - \theta_r)^{2-\frac{1}{\lambda}+2\lambda} \frac{\partial \theta}{\partial x_i} \right)$
$\qquad + \theta_e^{-3-2\lambda} \frac{\partial}{\partial x_3} \left( K_{s,3}(\bar{x})(\theta - \theta_r)^{3+2\lambda} \right)$  ;0< $\alpha$ <1 ; $\bar{x} = (x_1, x_2, x_3)$  .          (51)
Equation (51) shows that the fractional soil water flow equation (49) which accounts for the
time-dependent diffusivity expression of Guerrini and Swartzendruber (1992) and El Abd and
Milczarek (2004), does have an equivalent form where the integer time derivative of the soil
water content in the conventional Richards equation is replaced by a product of the fractional
time derivative of the soil water content and a fractional power of time, thereby supporting Sun
et al.'s (2013) conjecture, although in this study the fractional derivative is defined in the Caputo
sense while in Sun et al. (2013) the fractional derivative is defined with respect to a fractal ruler
in time.



In conclusion, in this study first a dimensionally-consistent continuity equation for soil water
flow in multi-fractional, multi-dimensional space and fractional time was developed. For the
motion equation of soil water flow, or the equation of water flux within the soil matrix in multi-
fractional multi-dimensional space and fractional time, a dimensionally consistent equation was
also developed. From the combination of the fractional continuity and motion equations, the
governing equation of transient soil water flow in multi-fractional, multi-dimensional space and
fractional time was then obtained. It is shown that this equation approaches the conventional
Richards equation as the fractional derivative powers approach integer values. Then by the
introduction of the Brooks-Corey constitutive relationships for soil water (Brooks and Corey,
1964) into the fractional transient soil water flow equation, an explicit form of the equation was
obtained in multi-dimensional, multi-fractional space and fractional time. Finally, the governing
fractional equation was specialized to the cases of vertical soil water flow and horizontal soil
water flow in fractional time-space. It is shown that the developed governing equations, in their
fractional time but integer space forms, show behavior consistent with the previous experimental
observations concerning the diffusive behavior of soil water flow.

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





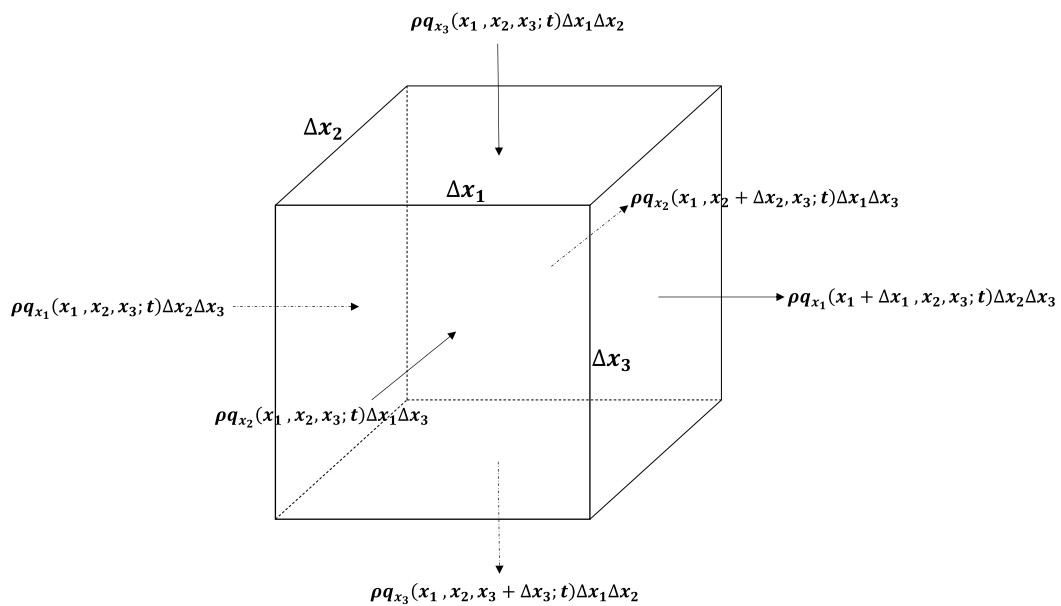

**Figure 1.** The control volume for the three-dimensional soil water flow