# Peer review of "Governing equations of transient soil water flow and soil water 1 flux in multi-dimensional fractional anisotropic media and 2 3 fractional time 4 5678"

_Hydrology and Earth System Sciences, 2016_

## Referee Comment (RC1) · Anonymous Referee #1 · 9 Nov 2016

This study derived the multi-dimensional fractional governing equation for soil water flow (Richards equation). The consistency check of dimension in the fractional governing equation is useful for application of this theory. I appreciate the effort on new mathematical techniques applied to hydrological governing equations. Therefore, I believe that this study is worth to be published in this journal. Followings are my additional questions/requests to the authors: It is interesting to see the some solutions of this equation, which might characterize some peculiar water flow patterns in soil matrix, if you have any. I am also curious that the physical meanings of the fractional governing equation. If the authors have any idea or even speculation about them, it would be very interesting to hear.

---

## Referee Comment (RC2) · T. Yamada (Referee) · 6 Jan 2017

T. Yamada (Referee)

yamada@civil.chuo-u.ac.jp

The present paper derived the continuity and motion equations in fractional time-space for unsaturated soil water. The deriving process is concise and convincing. It is very impressive that by simply replacing the original Taylor series to the fractional Taylor series in the deriving process, a dimensionally consistent fractional govern equations can be developed. Besides the new fractional govern equations is consistent with the fractional power number in the formulation of the diffusion coefficient very well. And the deriving process seems to be able to generalize to all similar problems. Thus, I believe the paper is qualified to be published in the journal. However, it would be better if the authors discuss the followings in detail. 1. As the paper mentioned, fractional

differential equation is an important approach to explain the non-Fickian dispersion in transport phenomena. It would be helpful if the authors give some details about whether the new govern equations can simulate the dispersion well or not. 2. The authors suggested that some former work had been done in the same topic like He(1998), but He's govern equation is not dimensionally-consistent. It would be interesting if the authors explain the difference between their work and He's, and how they solve the dimensionally-consistent problem.

---

## Author Comment (AC1) · 20 Jan 2017

**RESPONSE TO COMMENTS OF REVIEWER #1 ON HESS-2016-456 "GOVERNING EQUATIONS OF TRANSIENT SOIL WATER FLOW AND SOIL WATER FLUX IN MULTI-DIMENSIONAL FRACTIONAL ANISOTROPIC MEDIA AND FRACTIONAL TIME" by M. L. Kavvas et al.**

The authors thank Reviewer #1 for the insightful and constructive comments. Responses to the particular issues that are raised by the reviewer are given below.

REVIEWER #1:

The Reviewer #1 comments "I am also curious that the physical meanings of the fractional governing equation. If the authors have any idea or even speculation about them, it would be very interesting to hear."

Authors' Response:

The physical meaning of the fractional governing equation may be explained most easily in the case of vertical soil water flow. In the context of upstream-to-downstream vertical soil water flow from the soil surface downward, in the integer form of the soil water flow mass conservation equation (the traditional equation) the time rate of change of mass within the control volume grid (i) is determined by the mass flux coming from the upstream neighbour grid (i-1) into (i), and the mass flux that is moving from the control volume grid (i) to the downstream neighbour grid (i+1). This framework holds also for the soil water flow in the two horizontal directions. As such, the mass evolution in the case of the integer governing equation of soil water flow is local (at the scale of the specific computational grid), only due to interaction among neighbouring computational grids. On the other hand, in the case of the fractional governing equation of mass of vertical upstream-to-downstream soil water flow from the soil surface downward, the time rate of change of mass within the control volume grid (i) is due to fluxes of mass coming not only from the immediate nearest upstream neighbour grid (i-1) but from all possible upstream grids (i-1, i-2, i-3,…) (representing the intervals (x-$\Delta$x, x), (x-2$\Delta$x, x-$\Delta$x), (x-3$\Delta$x, x-2$\Delta$x)…) into control volume grid (i). As such, the fractional governing equations of soil water flow are nonlocal. In fact, in our paper the Caputo fractional derivative

$$\frac{\partial^{\beta} f(x)}{(\partial x)^{\beta}} = D_0^{\beta} f(x)$$

that is used for the fractional derivatives in the fractional governing equations of soil water flow in fractional time-space, is defined by (Odibat and Shawagfeh, 2007;  Momani and Odibat, 2008; Podlubny, 1999),

$$D_0^\beta f(x) = \frac{1}{\Gamma(1-\beta)} \int_0^x \frac{f\grave{}(\xi)}{(x-\xi)^\beta} d\xi \qquad 0 < \beta < 1, \quad x \geq 0.$$

As such, the Caputo fractional derivative superimposes nonlinearly (with weights $(x - \xi)^{-\beta}$ ) the effects of the local derivatives $f\grave{}(\xi)$ at each location $\xi$ in the interval (0,x) to the location x. Within this  framework, for example, in the case of one-dimensional downward vertical soil water flow in fractional time-space, the effect of  the upstream boundary condition is still accounted for at any depth x below the soil surface by means of the fractional spatial derivatives that appear in the corresponding governing equation (Equation (39) or Equation (46) in the paper).

REFERENCES:

Momani, S. and Odibat, Z.: A novel method for nonlinear fractional partial differential equations: Combination of DTM and generalized Taylor's formula, J. of Computational and Applied Mathematics, 220, pp.85-95, 2008.
Odibat, Z.M., and Shawagfeh, N.T.: Generalized Taylor formula, Appl. Math. Comput., 186(1), 286–293, 2007.
Podlubny, I.: *Fractional Differential Equations*, Academic Press, San Diego, 340pp, 1999.

---

## Author Comment (AC2) · 20 Jan 2017

**RESPONSE TO COMMENTS OF REVIEWER #2 ON HESS-2016-456 "GOVERNING EQUATIONS OF TRANSIENT SOIL WATER FLOW AND SOIL WATER FLUX IN MULTI-DIMENSIONAL FRACTIONAL ANISOTROPIC MEDIA AND FRACTIONAL TIME" by M. L. Kavvas et al.**

The authors thank Reviewer #2, Prof. T. Yamada, for his insightful and constructive comments. Responses to the two particular issues that are raised by him are given below.

1. " As the paper mentioned, fractional differential equation is an important approach to explain the non-Fickian dispersion in transport phenomena. It would be helpful if the authors give some details about whether the new govern equations can simulate the dispersion well or not."

Authors' Response:

In the context of solute transport in heterogeneous porous media Meerschaert, Benson, Baumer, Schumer and their co-workers (Meerschaert et al. 1999, 2002; Benson et al. 2000a,b; Baumer et al. 2005; Schumer et al. 2001), have shown by theoretical and numerical studies that the fractional advection-dispersion equation (fADE) which has fractional derivative powers both for the spatial derivatives and time derivatives in the governing equation of transport, has a nonlocal structure that can model well the heavy tailed non-Fickian dispersion. However, very few studies (mentioned in the paper) have addressed the underlying porous media flow in fractional time-space. The previous studies, cited in the paper, on the governing equations of soil water flow only treat time with fractional dimension, while keeping space with integer dimension. Furthermore, these previous studies address only one spatial dimension. Accordingly, our study attempts to develop a fractional continuity equation and a fractional water flux (motion) equation for unsaturated soil water flow in both fractional time and in multi-dimensional fractional space as the model for time-space fractional soil water flow that will provide the necessary flow information to the above-mentioned fADE non-Fickian model of transport in the case of soil media. At the moment we are working on the numerical application of the developed fractional soil water flow equations, and hope to complete this numerical application by the end of 2017. We hope that this numerical application will shed light into the question whether the dispersion can be simulated better by means of the new fractional governing equations, developed in our study.

2. "The authors suggested that some former work had been done in the same topic like He (1998), but He's govern equation is not dimensionally-consistent. It would be interesting if the authors explain the difference between their work and He's, and how they solve the dimensionally-consistent problem."

Authors' Response:

He (1998) was the first scholar who proposed a fractional form of Darcy's equation for water flux in porous media. Based on this fractional water flux equation, in his pioneering work He (1998) then proposed a fractional governing equation of flow through saturated porous media. However, the main objective of He's study was to develop a variational iteration method for the solution of fractional differential equations, which he successfully accomplished in his paper. Hence, he proposed a fractional form of the water flux in porous media rather than deriving it. He (1998) expressed the fractional Darcy flux as:

$$q_i(\overline{x},t) = -K_i(\overline{x},\theta)\frac{\partial^{\beta_i}h(\overline{x},t)}{(\partial x_i)^{\beta_i}} \qquad , 0 < \beta_i < 1 , i = 1,2,3 \qquad (1)$$

where $q_i$ is the water flux within the porous medium in the i-th direction (i=1,2,3), $K_i$ is the hydraulic conductivity of the porous medium in the i-th direction, h is the hydraulic head in the i-th direction, $x_i$ is the displacement in the i-th direction, and $\beta_i$ is the fractional power of the hydraulic gradient in the i-th direction. Denoting the length dimension by L, and the time dimension by T, for i = 1,2,3, the dimension of $q_i$ is L/T, the dimension of $K_i$ is L/T, and the dimension of h is L. Hence, the fractional derivative $\frac{\partial^{\beta_i}h(\overline{x},t)}{(\partial x_i)^{\beta_i}}$ of the hydraulic head h has the dimension of $L^{1-\beta_i}$, that is, length to the power (1- $\beta_i$), i=1,2,3. As such, the dimension of the left-hand-side (LHS) of Eqn. (1) is L/T while the dimension of the right-hand-side (RHS) of Eqn. (1) is $L^{2-\beta_i}/T$, for i = 1,2,3. The fractional soil water flux equation that was derived in our study is in the form:

$$q_i(\overline{x},t) = -K_i(\overline{x},\theta)\frac{\Gamma(2-\beta_i)}{x_i^{1-\beta_i}}\frac{\partial^{\beta_i}h(\overline{x},t)}{(\partial x_i)^{\beta_i}}, \qquad 0 < \beta_i < 1 , i = 1,2,3 \qquad (2)$$

Performing a dimensional analysis on our soil water flux equation (2), one obtains for the left and the right hand sides of the equation respectively,

$$[q_i(\bar{x}, t)] = L/T \qquad \text{and} \qquad \left[K_i(\bar{x}, \theta) \frac{\Gamma(2-\beta_i)}{x_i^{1-\beta_i}} \frac{\partial^{\beta_i} h}{(\partial x_i)^{\beta_i}}\right] = \frac{L}{T} \frac{L}{L^{1-\beta_i} L^{\beta_i}} = \frac{L}{T} \quad .$$

At the moment we are working on the numerical application of the developed fractional soil water flow equations, and hope to complete this numerical application by the end of 2017. We hope that this numerical application will shed light into the applicability of the developed fractional equations to the solution of soil water flow problems in fractional time and multi-fractional soil space.

REFERENCES:

Baumer, B., Benson, D. and Meerschaert, M.M.: Advection and dispersion in time and space, Physica A, 350, pp. 245-262, 2005.

Benson, D.A., Wheatcraft, S.W., Meerschaert, M.M.: Application of a fractional advection-dispersion equation, Water Resour. Res., 36(6), pp. 1403-1412, 2000a.

Benson, D.A., Wheatcraft, S.W., Meerschaert, M.M.: The fractional-order governing equation of Levy motion, Water Resour. Res., 36(6), pp. 1413-1423, 2000b.

He, J-H.: Approximate analytical solution for seepage flow with fractional derivatives in porous media, Comput. Methods Appl. Mech. Engrg. 167: 57-68, 1998.

Meerschaert, M.M., Benson, D.A., Baumer, B.: Multidimensional advection and fractional dispersion, Phys. Rev. E, 59(5), pp.5026-5028, 1999.

Meerschaert, M.M., Benson, D.A., Scheffler, H.P., and Baumer, B.: Stochastic solution of space-time fractional diffusion equations, Phys. Rev. E, 65, pp.041103-1 − 04113-4, 2002.

Schumer, R., Benson, D.A., Meerschaert, M.M., and Wheatcraft, S.W.: Eulerian derivation for the fractional advection-dispersion equation, J. Contam. Hydrol, 48, pp.69-88, 2001.

---

## Author Response (AR1)

**AUTHORS' RESPONSE TO THE EDITOR**

The authors thank the editor for his comment and his review decision.

As recommended by the Editor, the authors have revised the paper according to the comments of the reviewers.